# Prevalence of Early Marriage and Its Underlying Causes in Nepal: A Mixed Methods Study

Prakash C. Bhattarai [1,*], Deepak R. Paudel [2], Tikaram Poudel [3], Suresh Gautam [1], Prakash K. Paudel [1], Milan Shrestha [4], Janes I. Ginting [5] and Dhruba R. Ghimire [6]

1   Department of Development Education, Kathmandu University School of Education, Lalitpur 44700, Nepal;
    sgautam@kusoed.edu.np (S.G.); prakashpaudel@kusoed.edu.np (P.K.P.)
2   School of Business, Pokhara University, Pokhara 33700, Nepal; deepakpaudel@pusob.edu.np
3   Department of Language Education, Kathmandu University School of Education, Lalitpur 44700, Nepal;
    tikaram.poudel@kusoed.edu.np
4   Faculty of Education, Tribhuvan University, Kirtipur 44600, Nepal; milanshrestha313244@gmail.com
5   World Vision International Nepal, Lalitpur 44700, Nepal; janes_ginting@wvi.org
6   Independent Researcher, Kathmandu 44600, Nepal; dhruba.raj.gh@gmail.com
*   Correspondence: prakash@kusoed.edu.np

**Abstract:** Early marriage is one of the major traditional practices that affects the life of both boys and girls in many different ways. In this context, this research assessed the prevalence of early marriage and derived its underlying causes. Adopting a mixed methods approach, first, the study surveyed a sample of 1350 households of Nepal in which at least a marriage took place within the five years before the survey. Following a survey, secondly, the study explored 30 unique cases of early marriage, and ten among them were studied in more depth through face-to-face interviews. Logistic regression was applied to determine the factors that could influence the prevalence of early marriage. It was then followed by an analysis of the qualitative data. The research findings demonstrate that there is a high prevalence of early marriage (49.6%) among households within Nepal; nevertheless, the overall trend of early marriage is noted at a decreasing trend over the years. Undoubtedly, factors such as the level of education of the family members, the gender of the head of the household, and religion, influence the predisposition to early marriage within Nepal. Early marriage is undeniably a subjective phenomenon; however, such subjectivity is shaped by the socio-economic situation, as well as individual and family values. Thus, among others, the study implied that improving the strategies that promote higher formal schooling could reduce the prevalence of early marriage and thereby result in associated beneficial welfare effects in Nepal.

**Keywords:** determinants; early marriage; Nepal; prevalence

## 1. Introduction

Marriage is a union of two bodies, where a social, physical, and mental relationship is established (Little et al. 2014)—it is a union that demands rights, respect, and responsibility from each partner. In fact, marriage is an important milestone in the life of every human being (Haas and Whitton 2015), it is an imperative event predominantly guided by cultural social practices and particular customs associated with each culture (Berscheid 1998). These customary and social practices related to marriage have been institutionalized for generations (Ghimire and Samuels 2014).

In many cases, the inherited social customs contribute to the practices of early marriage and obliterate potential opportunities for social-economic progression for young boys and girls (Parsons et al. 2015). Globally, there are more than 60 million children who are affected by premature marriage (Arthur et al. 2018). What's more, Nepal is no exception, early marriage is prevalent across the country.

Nepal is a socio-culturally diverse country, with different castes, ethnicities, religions, and geographical areas (Central Bureau of Statistics (CBS) 2012). As a consequence of these diversities, Nepali people adopt a diverse set of rituals and practices that have both positive and negative consequences. Early marriage is an example of a practice with negative consequences. In fact, the practice of early marriage is a traditional practice that is more widespread amongst less advanced communities and this practice invariably affects the physical, intellectual, psychological, emotional, and educational aspects of those individuals who are engaged in the early marital practices (Bezie and Addisu 2019).

The practice of early marriage is a global concern (Arthur et al. 2018; Parsons et al. 2015). Furthermore, it is well documented that the consequences of marrying earlier are associated with health risks, child malnutrition, teenage pregnancy, access to education and even dropping out of school (Hotchkiss et al. 2016). Furthermore, marrying early limits children's choices in terms of their changing course of life and exposes them to a significant risk of abuse and violence (Bhandari 2019). Within Nepal, the marriage of one or both spouses under 20 years of age is defined as early marriage (Nepal Law Commission (NLC) 2017) and it is generally considered a violation of basic human rights globally (Bhandari 2019). Within Nepal, there are various laws such as the Constitution of Nepal 2015 to ensure that children's rights are upheld (Constituent Assembly Secretariat of Nepal (CASON) 2015). There is also the Nepal Civil Code Act 2017; this act also strongly prohibits marriage under the age of 20 years (Nepal Law Commission (NLC) 2017). The National Policy for Children 2013 (Ministry of Women 2016) also holds an objective of ending early marriage. Alongside these regulations, there is also the Children's Act 2018 (Nepal Law Commission (NLC) 2018) which also highly focuses on the need for a joint effort from government agencies, development partners, and other relevant working partners at the community level, to curb the social menace of early marriage.

However, the reality of curbing early marriage is far from straightforward. For example, Nepal is ranked as being the 16th highest with regards to early marriage practices around the world and the third-highest in South Asia (United Nations International Children's Emergency Fund (UNICEF) 2019). Further, as indicated by the last census 2011, 41% of the girls within Nepal become married before the age of 18 and 8.1% of these girls are pregnant between the ages of 15–19 (Central Bureau of Statistics (CBS) 2012). The Ministry of Health and Population (2011) reported that 55% of girls got married at the age of 18 and 74% of these girls were below the age of 20 years of age. This indicates the perpetuating scenario of early marriage in Nepal.

Within Nepal, early marriage is generally pervasive among the uneducated, *Janajatis* (Indigenous ethnic groups), *Dalits* (untouchable caste group), Muslims, *Madhesi* (inhabitants in the Terai regions of Nepal), and marginalized communities (Gautam 2019). Nonetheless, it deprives children of schooling, leading to mass unemployment, low income, and finally poor livelihood, as they evidently lack vocational skills and career development opportunities (McCleary-Sills et al. 2015). This immoral social practice also produces perilous results, in that it physically and psychologically harms young girls by causing physical risks to reproductive organs, unwanted sexual intercourse, pregnancy, fear of sexual brutality, transmitting sexual diseases, and maternal mortality (Goli et al. 2015). Therefore, this research contributes to the field by assessing the present situation of early marriage in the country and also investigates the causes behind the prevalence of early marriage. Although the study may not represent the whole country, it gives a scenario that has been an important knowledge contribution to plan strategies for reducing the incidence of early marriage among Nepali youth.

## 2. Literature Review

The right to free will and consent with regard to marriage is recognized in the Universal Declaration of Human Rights (UDHR), as it specifies that consent cannot be free and full, when one is not sufficiently matured, to make an informed decision about a marital partner. In fact, the adverse impacts of early marriage on the lives of children, have been increasingly

documented within global studies (Arthur et al. 2018; Bajracharya and Amin 2012; Bezie and Addisu 2019). Early marriage inevitably impacts the child's health conditions—including physical, psychological, emotional, sexual and reproductive—as well as the social and economic aspects of their lives (Raj 2010). More importantly, there also exist gender-specific differences that need to be noted concerning early marriage. Where, girls are substantially at greater risk of being exposed to early marriage and the girls who do become mothers below 20 years of age, have a 50% higher rate of pre-natal and infant mortality, morbidity, and newborn deaths, in comparison to women who become mothers after 20 years of age (World Health Organization 2011). More specifically, girls in the Asia Pacific region are also more likely to experience domestic violence or other forms of abuse as they are culturally perceived as a financial burden and, therefore, early marriage is seen as a solution to this (Bajracharya and Amin 2012).

The civil code of Nepal defines marriage as a state of acceptance as husband and wife by men and women through social practices or bondage with other legal rules (Nepal Law Commission (NLC) 2017). Before the General Code (the *Muluki Ain*) in 1963, marriage predominantly was a customary rule, marriage at an early age, especially among girls. Parents used to arrange the marriage of their daughters when they began the menstruation period and it was a preferred practice in Nepali societies. The code, for the first-time, legalized marriage on the basis of age (18 for women and 20 for men) on their parents' consent. Later Nepal Marriage Bill 2011 made 21 years for men, and later in 2017, the National Civil (Code) Act amended the minimum 20 years for marriage for both men and women. The rationale for 20 years is that both the boys and girls at least complete their school education, become mature and independent, and can make a choice for themselves (Bhandari 2019).

Pregnancy even at a young age is not a social stigma if the pregnant girl is married with customary rules. The National Civil (Code) Act 2017 legalizes a baby to its father and mother (even they are under the legal age of marriage) although it does not legalize their marriage. However, the code permits registering marriage after the parents of a child reach the legal age. According to the National Penal (Code) Act 2017, both marriages under legal age and rapes are offensive activities (Ministry of Law, Justice, and Parliamentary Affairs 2017). The act treats both cases as a crime and has a strong penalty such as imprisonment. Sexual relations under the age of 18, even with consent legally treated as rapes. However, such as case can be exempted if no one files a case against such an activity to a law court.

The practice of early marriage is prevalent amongst all the varying ethnic groups and castes in Nepal (Gautam 2019). However, it prevails more so in poor and socially excluded groups (Central Bureau of Statistics (CBS) 2014). Some concentrated efforts were made, against early marriage in Nepal, largely due to the damaging effects of early marriage relating to the physical and mental health of the participants of it. For example, as a signatory nation of the United Nations Convention on the Rights of the Child (1989), the Government of Nepal has a Ministry of Women (2016), which aims at meeting Sustainable Development Goals (SDG) indicators and ending early marriage by the year 2030 (Ministry of Women 2016). The legal code of Nepal 2017 mentions marriage under the age of 20 is an offensive act even if it happens with their parental consent. It provisions penalties for everyone who breaches the legal codes and registration section of local bodies that registers marriage considers it as illegal. Hence, such marriage cases are not authorized.

There have also been initiations from local governing bodies within Nepal, such as the operationalization of the National Child Rights Council (2019). However, despite these numerous efforts, Nepal has still been reported as having one of the highest rates of early marriages in Asia—for both girls and interestingly even boys (United Nations Children's Fund (UNICEF) 2017).

No doubt, religion, is often embedded within the customary and social practices of many cultures and countries (Bhattarai 2009). However, religion provides particular mandatory rules, concerning the practices of marriage. Roux and Palm (2018) argue that religion has direct and indirect impacts on marriage practices. For example, the religious

codes become powerful rituals, among the followers and often decide the marital fate of the sons and daughters of many (Singh 2008). However, Cader (2017) interestingly argued that marriages at an early age continue, because parents encourage the premature marriage of their daughters to protect them from any unwanted and unfortunate happenings in their life.

The practice of child marriage is common among economically vulnerable communities (Gautam 2019). In fact, it is well documented that parents in economically vulnerable communities are hesitant to invest in the education and health of their daughters (McCleary-Sills et al. 2015). Instead, they may prefer to invest in other sectors that ensure more immediate financial gains for the family or their sons. Conversely, Parsons et al. (2015) have argued that in some communities, the bride's family also acquires an immediate economic benefit from the bridegroom's family. Many parents prefer marrying off their daughters at an early age because they have to pay less money in the form of a dowry and such social norms have created gender stereotypes (Roux and Palm 2018), that consider early marriage a complex phenomenon.

This shows that early marriage in the South Asian context such as Nepal is still a threat in many cases such as economic, social, psychological, etc. to both boys and girls. Some initiatives are apparent at the policy level nationally as well as internationally, however, the prevalence of early marriage shows there has not been a subsequent achievement in real practice. Against this backdrop, this paper becomes a crucial document that predicts the underlying causes of perpetuating early marriage cases in Nepal.

## 3. Methods

The study was conducted in seven districts (Achham, Bajhang, Doti, Kailali Rautahat, Mahottari, and Sarlahi) from two provinces of Nepal. However, the study also covered the Kanchanpur and Dolakha districts as adjunct research areas, as these locations share similar socio-cultural settings. The rationale for selecting these districts was: the higher incidence of child marriage (Central Bureau of Statistics (CBS) 2014) and the geographical diversity within the areas.

The study followed mixed methods research (Creswell and Clark 2018), where the researcher concurrently collected and analyzed the data and then integrated the findings, drawing inferences using both qualitative and quantitative approaches and methods (Teddlie and Tashakkori 2009, p. 14). We firstly conducted a community-based cross-sectional household survey with a sample of 1350 households in 2020 that represented the major religious and socio-economic groups of Nepal. In order to capture the possibility of recent incidence of early marriage, we only approached the households in which marriages took place within the five years prior to the survey.

The households were randomly selected from the cluster, to meet the sampling requirements. The five-stage sampling procedure was used starting from the province, district, local level, ward or cluster, and household. This included a total of 34 local levels out of 753 across the country. The households were selected based on a first come first served criteria, across the radius of the location of the ward's office, as per the 2017 restructuring of administrative divisions of Nepal after the new constitution which was promulgated in 2015.

The sample size was determined, using Cochran's corrected formula (Bartlett et al. 2001) by applying multistage cluster sampling. The calculation of the sample size (*n*) was based on a single proportion formula and has been expressed in Equation (1).

$$n = \left[ \frac{z^2 P\,(1-P)}{e^2} \times deff \times NR \right] + Additional\ Adjustment\ Sample \qquad (1)$$

Within the study, 50% of the proportion of success of a key indicator (*P* = 0.5) was considered, with a 95% desired level of confidence (*z* = 1.96), assuming a 5% margin of error (*e* = 0.05), and the value of design effect as 2.375 (*deff* = 2.375) that generally ranges from 1.25 to 3.50 (Shackman 2001). Furthermore, the most common 10% non-response rate

(*NR* = 1.10) for the household survey (United Nations (UN) 2008) was considered and for these 347 additional adjustment samples were added.

Onwuegbuzie and Johnson (2006) propose using the same population to address one of the major concerns of mixed methods research's validity. In this regard, the researchers selected 30 cases of early marriage; five from each cluster and interviewed 87 participants including ritual experts, members of child clubs, members of women groups, individuals of government/non-government agencies, and boys and girls who married early from the households that participated in the survey. The cases were selected from those households which were either economically vulnerable or exemplary considering the prevalence of early marriage. The researchers explored the cases with participants' voluntary consent in detail. This method was imperative because, as McKim (2017) suggests, the integration of quantitative and qualitative data, increases the value of mixed methods research because the blending leads to a greater understanding of the issue (Bryman 2006; Yin 2018). In this study, the researchers also integrated findings of quantitative research, with cases of qualitative research as this assisted them in providing a comprehensive understanding of predicting factors of early marriage and how it prevailed in the real context.

### 3.1. Variables

The dependent variable within this study was a binary outcome of early marriage and took the value of 1 for households that engaged in early marriage and 0 for households that did not engage in early marriage. The independent variables include different characteristics, such as individuals, households, community factors as suggested by prior research (Bezie and Addisu 2019). Age was categorized into three levels, (1) 20–39 years (Young), (2) 40–59 years (Youth) and (3) 60 and more (Old), as suggested by the National Youth Policy (Ministry of Youth and Sports 2015). However, originally, the age variable was divided into 7 categories such as 20–29, 30–39, 40–49, 50–59, 60–69, and 70 and more. The economic status of the household was constructed by considering the various income sources of the household, such as agriculture, government sources, labor, business, wages, foreign employment, social security funds, pension and so on. However, in terms of descriptive analysis categories, households were categorized into poor (below the average national income) and not poor (above the average national income).

### 3.2. Instrumentation

The survey questionnaire was developed to assess the socio-cultural norms, economic and social vulnerabilities affecting early marriage. First, the authors developed the scale in consultation with the experts, i.e., ritual experts, government officials, academicians, and desk-based review. Then, the developed questionnaire was translated into Nepali from the English language and again in the Nepali version for ensuring the expected sense. Similarly, developed guidelines for qualitative interviews were based on themes of the survey questionnaire to explore the present situations, prevalent practices, patterns of early marriage, and identify measures, strategies, and plans to tackle early marriage practices. However, the researchers were free to generate and ask other related questions to capture in-depth information.

### 3.3. Data Management and Analysis

The researchers managed and interpreted the obtained data from the survey and qualitative interactions concurrently. The survey data was exported into SPSS version 26 (IBMers, New York, NY, USA), from KOBO Toolbox (Kobo Inc., Toronto, ON, Canada) for analysis while qualitative information was coded, thematized, and generated meanings employing MXQDA version 20 (VERBI GmbH, Berlin, Germany).

The first sample characteristics of study participants that engaged in early marriage were presented employing descriptive statistics as univariate analysis. Furthermore, this study categorized, the obtained responses taking into consideration individual attitude

and role of religious leaders towards early marriage into three levels by employing the strategy of Best (Shrestha 2019):

$$\frac{Higher\ score - Lower\ score}{Number\ of\ Levels} = 1.33 \tag{2}$$

More specifically, the obtained mean score was categorized into three levels using the interval of 1.33. They were negative, neutral, and positive, in terms of attitude and low, moderate, and high, concerning the impact of religious leaders towards early marriage, in the category of 1.00–2.33, 2.34–3.66, and 3.67–5.00, respectively. Secondly, cross-tabulation analysis was carried out, between each of the socio-economic factors and early marriage, using Pearson's $\chi^2$ as bivariate analysis. Finally, logistic regression was applied, to find the most prominent determinants of early marriage as a multivariate analysis. Before logistic regression was applied, the multicollinearity among the independent variables was assessed. The fitted model displays, the estimated adjusted odds ratios (ORs), along with a 95% confidence interval (CI).

In the Mixed Methods Research, a researcher 'collects and analyzes data, integrates the findings, and draws inferences using both qualitative and quantitative approaches or methods in a single study' (Teddlie and Tashakkori 2009, p. 14). Considering this typology, the researchers integrated survey results with qualitative interviews. Doing so was also a method to a greater understanding of the issue (Bryman 2006; Yin 2018). For this, the researchers explored the underpinning causes of early marriage by interacting with various stakeholders, particularly boys/girls who married early, cultural leaders, political leaders, and government officials once the insight derived from the integration. We considered the sociocultural aspect to be a factor in the case of early marriage. The researchers assigned aliases to the research participants within this study and they voluntarily participated with their consent in this research process.

## 4. Prevalence of Early Marriage: Findings from the Survey

The statistics demonstrated that, overall, 670 of 1350 (49.6%) households studied engaged in early marriage, in the five years prior to the survey. This revealed that there is a prevalence of early marriage in Nepal. Nevertheless, the study also observed that marriage below 20 years of age is at a decreasing trend in the last five years. More specifically, the rate of the households whose sons/daughters marry below 20 years in 2015 was 51.8%. In 2019, this rate came down to 43.7% showing a trend of gradual decrease. The trend of decrement in early marriage is also seen in the national data published by (Central Bureau of Statistics (CBS) 2014) in Nepal where the percent of child marriage below 15 years were 5.7, 1.3, and 0.8 in 1991, 2001, and 2011, respectively. Among the early marriage cases, nearly half, i.e., 47.1% incurred at the age group of 15–19 and 3.84% married below 15 years.

Among the surveyed households, the majority (72%) were male-headed households (Table 1). Nearly two in five heads (42%) were 40–49 years of age, followed by 21% who were 50–59 years of age. More than half (55%) were Hindus, and 18% were Buddhists. Nearly one-fourth (24%) of the respondents did not have a formal education. The majority of the respondents (65%) had a negative attitude towards early marriage and 57% of the respondents held the perception that religious leaders had a small role to play in early marriage. Table 1 depicts the characteristics of the variables included in this study.

**Table 1.** Sample Descriptive Statistics.

| Characteristics | Categories | Number | Percent |
|---|---|---|---|
| Gender of head | Male | 978 | 72.4% |
| | Female | 372 | 27.6% |
| Age of head in years | 20 to 39 | 397 | 29.4% |
| | 40 to 59 | 844 | 62.5% |
| | 60 or more | 109 | 8.1% |
| Religion | Hindu | 742 | 55.0% |
| | Buddhist | 240 | 17.8% |
| | Muslim | 188 | 13.9% |
| | Christianity | 120 | 8.9% |
| | Kirant | 60 | 4.4% |
| Family type | Joint | 699 | 51.8% |
| | Nuclear | 651 | 48.2% |
| Migration status | Not migrated | 1192 | 88.3% |
| | Migrated | 158 | 11.7% |
| Highest level in of education in family | Illiterate | 143 | 10.6% |
| | Literate | 178 | 13.2% |
| | Basic | 306 | 22.7% |
| | Secondary | 523 | 38.7% |
| | Bachelor (College) | 200 | 14.8% |
| Household economic status | Poor (Below average level) | 1009 | 74.7% |
| | Not poor (Above the average level) | 341 | 25.3% |
| Locale | Urban | 858 | 63.6% |
| | Rural | 492 | 36.4% |
| Attitude towards early marriage | Negative | 871 | 64.5% |
| | Neutral | 249 | 18.4% |
| | Positive | 230 | 17.0% |
| Role of religious leader toward early marriage | Low | 770 | 57.0% |
| | Moderate | 319 | 23.6% |
| | High | 261 | 19.3% |
| Total of each characteristic | | 1350 | 100.0% |

### 4.1. Early Marriage Due to Socio-Demographic and Related Characteristics

The socio-demographic characteristics such as gender, religion, migration status, and education, were significantly related to incurring early marriage. Interestingly, the female-headed households tended to engage with early marriage practices, more so than male-headed households. For example, the percentage of households experiencing early marriage was 57% among female-headed households, whereas it was only 47% among male-headed households. So, there was a difference of 10% and the result was statistically significant, at less than 1% level. The prevalence of early marriage was the highest among those who were Muslims and it was the least among those who followed the Christian religion in their households (62% vs. 33%). Migrated households had a comparatively lower rate of early marriage, compared to those households that had not migrated (39% vs. 51%). Where higher levels of education were prevalent in the family, there was a strong negative association with early marriage. For example, the prevalence of early marriage

was the highest among those households that were considered illiterate, whereas it was only 25% among those households, that had at least one family member with a bachelor's level of education.

The households that were classed as being low income were more likely to approve early marriage in comparison to their counterparts (51% vs. 47%). However, the difference was not statistically significant, within a 5% level of significance. The distribution of incidence of early marriage according to locale, attitude and role characteristics have also been presented in Table 2. Interestingly, living in a rural area was also an important contributing factor that often led to early marriage. For example, 54% of rural households experienced early marriage, whereas the ratio in urban areas was 47%. The incidence of early marriage was in fact, the highest among those who had a positive attitude towards early marriage (75%), compared to those who had a negative attitude towards early marriage (39%).

**Table 2.** Early Marriage due to Socio-demographic and Related Characteristics.

| Characteristics | Categories | Presence (Incidence) of Early Marriage | | Chi-Square Value |
| --- | --- | --- | --- | --- |
| | | No (%) | Yes (%) | |
| Gender of head *** | Male | 53.0 | 47.0 | 9.56 |
| | Female | 43.5 | 56.5 | |
| Age of head in years | 20 to 39 | 49.4 | 50.6 | |
| | 40 to 59 | 49.9 | 50.1 | 2.64 |
| | 60 or more | 57.8 | 42.2 | |
| Religion *** | Hindu | 44.1 | 55.9 | |
| | Buddhist | 68.3 | 31.7 | |
| | Muslim | 37.8 | 62.2 | 71.48 |
| | Christianity | 66.7 | 33.3 | |
| | Kirant | 63.3 | 36.7 | |
| Family type | Joint | 51.6 | 48.4 | 0.94 |
| | Nuclear | 49.0 | 51.0 | |
| Migration status ** | Not migrated | 48.9 | 51.1 | 8.69 |
| | Migrated | 61.4 | 38.6 | |
| Highest level of education in the family *** | Illiterate | 23.1 | 76.9 | |
| | Literate | 45.5 | 54.5 | |
| | Basic | 43.5 | 56.5 | 103.29 |
| | Secondary | 53.9 | 46.1 | |
| | Bachelor (College) | 75.5 | 24.5 | |
| Household economic status | Poor (Below average level) | 49.4 | 50.6 | 1.64 |
| | Not poor (Above average level) | 53.4 | 46.6 | |
| Locale * | Urban municipality | 52.6 | 47.4 | 4.53 |
| | Rural municipality | 46.5 | 53.5 | |
| Attitude towards early marriage *** | Negative | 61.5 | 38.5 | |
| | Neutral | 34.9 | 65.1 | 127.41 |
| | Positive | 24.8 | 75.2 | |
| Role of religious leader in early marriage ** | Low | 54.2 | 45.8 | |
| | Moderate | 48.9 | 51.1 | 13.86 |
| | High | 41.0 | 59.0 | |

*** $p < 0.001$, ** $p < 0.01$, * $p < 0.05$. $p$-value is based on chi-square statistic.

### 4.2. Factors Contributing to Early Marriage

Table 3 outlines several socioeconomic, community, and attitudinal factors that contributed to the prevalence of early marriage within Nepalese households. The logistic regression yielded a wide range of determinants linked with the prevalence of early marriage and the model accuracy rate of predictability was 69.56%.

The socioeconomic factors within this study found that the gender of the head of the household, religion and the level of education within the family, showed a strong association with marriage at an early age. In particular, female-headed households were 57% more

likely to engage in early marriage, compared to male-headed households, while controlling all other potential socioeconomic confounders (OR = 1.57, $p < 0.001$, 95% CI = 1.20–2.07).

**Table 3.** Determinants of Early Marriage, Results from Logistic Regression.

| Characteristics | Explanatory Variables | ORs | 95% CI for ORs | |
|---|---|---|---|---|
| | | | Lower | Upper |
| **Socioeconomic characteristics** | Gender of the head (Male = R) | 1.00 | | |
| | Female | 1.57 *** | 1.20 | 2.07 |
| | Head's age in years (20–39 = R) | 1.00 | | |
| | 40 to 59 | 1.05 | 0.80 | 1.37 |
| | 60 or more | 0.75 | 0.46 | 1.21 |
| | Religion (Hindu = R) | 1.00 | | |
| | Buddhist | 0.45 *** | 0.31 | 0.66 |
| | Muslim | 1.11 | 0.76 | 1.62 |
| | Christianity | 0.53 *** | 0.33 | 0.84 |
| | Kirat | 0.53 * | 0.29 | 0.99 |
| | Family type (Nuclear = R) | 1.00 | | |
| | Joint | 0.89 | 0.69 | 1.14 |
| | Migration status (Not migrated = R) | 1.00 | | |
| | Migrated | 0.98 | 0.67 | 1.45 |
| | Education level (Illiterate = R) | 1.00 | | |
| | Literate | 0.41 ** | 0.24 | 0.68 |
| | Basic | 0.60 * | 0.37 | 0.99 |
| | Secondary | 0.38 *** | 0.24 | 0.62 |
| | College | 0.13 *** | 0.07 | 0.23 |
| | Household annual income | 1.00 | 1.00 | 1.00 |
| | Urban locale | 1.25 | 0.95 | 1.64 |
| **Attitude and role characteristics** | Attitude towards early marriage (Negative = R) | 1.00 | | |
| | Neutral | 2.87 *** | 2.06 | 4.01 |
| | Positive | 3.74 *** | 2.55 | 5.48 |
| | Role of the religious leaders on early marriage (Low = R) | 1.00 | | |
| | Medium | 0.94 | 0.69 | 1.27 |
| | High | 0.95 | 0.67 | 1.34 |

*** $p < 0.001$, ** $p < 0.01$, * $p < 0.05$, R = Reference category, Model correctly classified (Overall percentage of model predictability) = 69.56%.

The religion of the head of the household appeared to be a significant factor in contributing to early marriage and the results revealed a significant relationship. Buddhists were 55% less likely to incur early marriage, compared to those who followed Hinduism as a religion in their households (OR = 0.45, $p < 0.001$, 95% CI = 0.31–0.66). Similarly, Christians were 47% less likely to engage in early marriage, compared to households following the Hindu religion (OR = 0.53, $p < 0.001$, 95% CI = 0.33–0.84). However, Muslims were not significantly more likely to have an early marriage, compared to Hindus.

Interestingly, the level of education within a family had a strong influence in terms of reducing marriage at an early age within Nepalese households. A higher level of education in the family was associated with a decreased tendency to marry at an early age (ORs = 0.41, 0.60, 0.38, 0.13 for the literate, basic, secondary, and college-level education, respectively). In fact, households with bachelor-level education were 87% less likely to marry at an early age when compared to households where all members were illiterate (OR = 0.13).

The household income, a proxy measure of the economic/poverty status of the household, did not reveal a significant association with early marriage at a 5% level of significance. Moreover, the quartile category of household income (very poor, poor, rich, and very rich) was also calculated and fitted into the logit model. The result was still insignificant, revealing that poverty was not a significant predictor of early marriage.

When considering the attitude towards early marriage, respondents who had a positive attitude were 3.74 times more likely to have a marriage at an early age, compared to those who had a negative attitude (OR = 3.74). Similarly, individuals who had a neutral attitude about early marriage were 2.87 times more likely to experience marriage at an early age, compared to those who had a negative attitude towards early marriage. However, it is important to note that attitude is a perceptional attribute, that could be changed over time. Therefore, this study assessed the current level of attitude of respondents, towards early marriage. Furthermore, variables such as head of families age, type of family, migration status, poverty/economic status, urban-rural locale and the role of religious leaders with regard to early marriage were not seen as contributing factors to marriage at an early age in multivariate logistic regression analysis.

Underlying Causes of Early Marriage: Socioeconomic or More?

The researchers of this study also engaged in qualitative inquiry to explore and better comprehend the underlying causes of early marriage. The data was themed and coded into broad categories of socio-cultural, and economic-related causes.

Firstly, we explored cases of early marriage considering the socio-cultural aspects as an underlying cause. Participants of the study claimed that there was a decreasing trend of early marriage, which was partially true with the result of the survey result. However, participants viewed the underlying causes vividly. The perceived causes were most common concerning the socio-cultural and economic context. Raju, a male local political leader, shared:

> "In my community, people take 'girls' as *pagadi* as a symbol of pride; if she elopes or runs away with someone, especially from other castes, parents lose their respect in society. If the boy does something, the matter is not taken as seriously as in the case of a girl. Hence, parents want their daughters to get married early so that they would not face such a situation."

Similarly, interactions with Radha, a female participant, who married in her teens revealed that it was her compulsion to become married as she could not discard her family rituals. She further shared:

> "I was married when I was 16. My parents decided on my marriage when my husband's father came to my home with Sagun (local wine and special food) with a marriage proposal. It was the first proposal and among my community people, they do not want to avoid the first proposal. They believe in *deuta risaune* 'God may get angry' if the proposal is rejected."

Such a compelling situation was also reported by another female participant who had to marry at an early age. She explained:

> "I am the 7th daughter of my parents. Before my marriage, my father was seriously ill. Once, a religious guru predicted that his illness might be cured if the 7th child got married. My parents were convinced and planned my marriage. At that time, I said nothing since it was related to my father's life. But nowadays, I regret accepting the proposal passively."

During the interaction with the participants, it was increasingly observed that early marriage was perceived as synonymous with girls' marriage. Most of the participants interpreted early marriage as if it was the phenomenon associated only with the girls within their community. Nevertheless, the early marriage cases were equally prevalent among the boys. Kiran, a male participant who married at an early age shared his views:

> "Whenever I used to be with my grandfather, he had always one request that I should marry. He was in his eighties and used to share whether he would die without seeing his granddaughter-in-law. Later, he started insisting that his death would not come easily until I married."

The researchers confronted with the fact of early marriage that it also prevailed among the males though participants were found understanding mostly associating with females. In fact, during the study, many cases were reported where males were either forced or unaware of early marriage. One such case was of Krishna, a male participant, who was forced to marry at an early age by his parents. The boy reminisced:

> "I could not avoid the pressure of my family when they forced me to marry early. It was a pressure created when my parents went to show my china (horoscope) with the religious guru (leader). The guru pointed out that I had to marry before 17 otherwise after 25. If I married in between, there might be misfortune even death. In this respect, my parents did not agree to marry lately, i.e., after 25."

Significantly, it was evident that male members of the family were able to contribute to their decision with regards to premature marriage, whereas females were not able to assert the same control over their marital fate. This is evidence of the dominating patriarchal values of Nepali society, where males are still perceived as the traditional and dominant breadwinners of the family. A boy who married early shared his views:

> "When I married, I didn't know about the legal age to be married clearly. Before marriage, I was in India and was studying there in grade 8. I saw the girl (now my wife) when I went to my aunt's (father's sister) house and was determined for marrying her since I found her most beautiful. I thought I had to marry her anyhow. So, I even quit school. In the beginning, I couldn't propose to her but later my aunt supported me to persuade her for the marriage."

Evidently, the socio-cultural aspects were key contributing factors that were pivotal in determining whether premature marriage occurred within these households, in Nepal. Indeed, people from vulnerable groups may marry earlier, but in some cases, girls from educated families also tend to become married at an early age. However, this finding was also corroborated by the survey findings and the cases of early marriage were observed where families had a lower level of education. The reason for this could be that the higher one's educational attainment, the more knowledge he/she acquires and understands, including all kinds of information about reproductive health, the best marriage age, and the effect of marrying at an early age (Bezie and Addisu 2019).

Secondly, this study sought to explore cases of early marriage, particularly taking into consideration the economic aspect as well. From the interviews, participants expressed their general view, that people wanted to become married after becoming economically independent. However, the income-generation process itself was challenging due to lack of education or qualifications, the competitive job market, or low-income jobs. Given this context, a local political leader argued, that the likelihood of marrying early is high. He shared his views: "Parents believe that if their idle son gets married, he would (be forced to) start earning."

The participants also associated one of the motivating factors behind early marriage as the dowry practice, which has been prevalent in most of the regions of Nepal for some time.

In fact, a female participant shared that within a joint family, a girl's parents would prefer, that their daughter married earlier rather than later. So that the expenses of the marriage are covered from common property. If the marriage takes place within a joint family (i.e., with the grandparents, father's brothers, and their families), the parents don't have to bear all expenses; it is the overall responsibility of the whole family, a matter of great relief for the girl's father. If the marriage takes place after all the brothers are separated (nuclear family), the father has to manage all the financial burdens on his own. Therefore, she or he may potentially not receive any financial help from his/her family members (brothers and parents if alive), whereas in a joint family there would be significant financial help available to them.

It was also evident that boys were likely to drop out of school after grade ten and search for jobs. Some of the boys even aspired to acquire a dowry, with the view to starting

up their own business and this was, therefore, their motivating factor for early marriage. In this sense, it is evident that the economic vulnerability of the family and the expectation of gaining economic benefit seems to contribute to marrying at an early age. Likewise, in the case of girls, the family feels relieved of the economic expenses while, in the case of boys, they feel they are adding to the labor force. However, importantly to a local political leader, it was neither a relief to the parents nor a beneficial addition to the labor force. In fact, a local political leader shared his objective views:

> "Instead of making a balance in the family many economic and physical challenges increase as their expenses increase and their children are born early."

This indicated the association between poor economic conditions and the occurrence of early marriage (Figure 1), and it is a vicious cycle from one generation to the next, and therefore an intervention was needed.

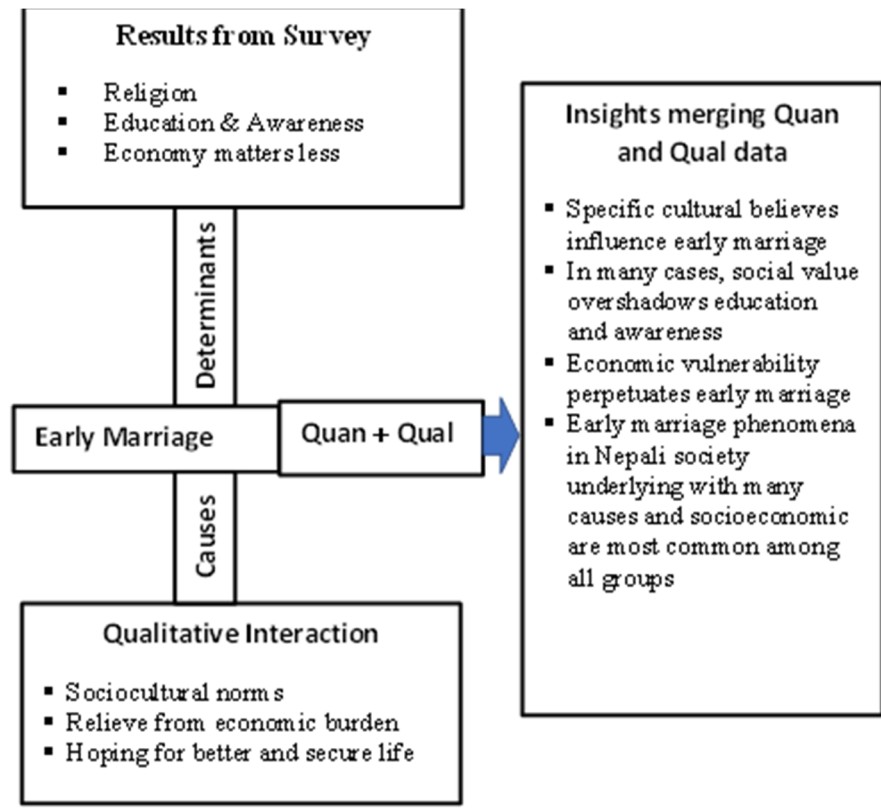

**Figure 1.** Joint Display: Merging Quantitative and Qualitative Results. Quan = quantitative strand; Qual = qualitative strand.

## 5. Discussion

Despite international regulation agreements and national laws, early marriage is found still highly prevalent within Nepal. This sort of ill practice can have significant physical, intellectual, psychological, and emotional effects. In fact, this practice results in fewer educational opportunities, and fewer opportunities for personal growth for both boys and girls (Bezie and Addisu 2019). The household survey of this study found that nearly half of the Nepali households experienced premature marriage. This finding is substantially higher than the figures previously highlighted in prior studies conducted in Nepal and elsewhere (Bezie and Addisu 2019; Guragain et al. 2017). The rationale behind the high level of early marriage could be because this study considered the household as the unit of analysis, whereas most prior studies have tended to consider individuals as the units of analysis. Furthermore, the high incidence of early marriage within these households, indicates that early marriage in Nepali households is still unacceptably very

high. Therefore, the governing and local bodies of Nepal need to consider this when designing social policies for the country.

The present findings of the study revealed that the educational level in the family had a significant association that incurs early marriage. The households headed by a female member and having no formal education likely to have more early age of marriage cases. Interestingly, households with college-level education were evidently well informed and therefore less likely to marry earlier, compared to households where all members were illiterate. This finding is consistent with the recent studies from Congo (Mpilambo et al. 2017), India (Paul 2020), Serbia (Hotchkiss et al. 2016), Sudan (Ali et al. 2014), and Nepal (Guragain et al. 2017). This could be linked with an individual's higher educational attainment resulting in more knowledge of the repercussions of early marriage (Bezie and Addisu 2019). However, evidently, if one's educational achievement is poor, there is a disconnection of knowledge and information. This, therefore, indicates the importance of education as a social phenomenon, particularly for early marriage. However, this study also explored cases of early marriage amongst households that were better educated. Participants perceived it to be an issue that had emerged as a result of strong sociocultural values, which in many cases compels people to marry earlier. However, it was evident that such marriages, were either forceful or were conducted with the hope for a better and financially secure life.

When considering the gender-related aspects of this study, it was evident that female-headed households were significantly more likely to encourage the practice of early marriage rituals, compared to male-headed households. The reason behind, the incidence of a higher level of early marriage, with female-headed households, could be such that social roles play a key role in determining gender equity in many developing countries (United Nations International Children's Emergency Fund (UNICEF) and United Nations Population Fund (UNFPA) 2018). Furthermore, Santow (1995) has abundantly documented the female disadvantage, in less developed countries about social well-being.

A significant relationship between religion and early marriage was revealed within this research study. This derived result is analogous to the study of Sayi and Sibanda (2018) where they established that there was a correlation between religion and early marriage. This research study found that Buddhists, Christians, and Kirants were significantly less likely to have an early marriage compared to the Hindus and Muslims in this study. The social well-being of family members is largely impaired by culturally determined roles of religion/ethnicity, through a complex web of physiological and behavioral interrelationships and synergies that permeate every aspect of their lives (Santow 1996). Thus, priority needs to be given to less advantaged religious groups such as Muslims and Hindus.

This study also revealed that individuals that had a positive attitude about early marriage, were significantly more likely to face early marriage compared to their counterparts. The social-psychological attitude and beliefs are a key influence on their children's behavior and the determinants of the appropriate age of marriage (Jennings et al. 2012). These deep-rooted cultural practices need significant efforts at the local, provincial, and federal levels to abolish them and reduce the practices of early marriage in Nepal. Variables such as the head of the households' age, type of family, migration status, poverty status, urban-rural locale, and the role of religious leaders in influencing early marriage, were not seen as contributing factors to marriage at an early age in statistical analysis. However, the contribution was noticeable in sampled households. This might also be because early marriage was a subjective phenomenon and largely influenced by sociocultural aspects. The households included in the study carried specific socio-cultural characteristics, that might be differently manifested in other cultures. In fact, it resulted differently when we generalized the findings in a large population.

The study shows the prevalence of marriage below 20 years among girls and boys. This also questions the target of achieving the global agenda of sustainable development, particularly the Sustainable Development Goal 5 (SDG 5: Gender Equality) which aims to eliminate all: "harmful practices, such as child, early and forced marriage and female

genital mutilation" (United Nations (UN) 2021). The prevalent early marriage practice in Nepal also has increased multi-facet challenges to achieve universal access to sexual and reproductive health and reproductive rights of girls. The national legal framework does not permit marriage below 20 years but it continues to exist and prevail more among the under-privileged groups of people. As a result, a serious threat to decision-making for the wellbeing of women affects gender equality in society.

## 6. Conclusions

No doubt that the prevalence of early marriage is relatively high in Nepal. The socio-economic factors that tend to assist early marriage in this study include the education level of family members, the gender of the household head and religion. Moreover, the social status and prestige of families also affect the practices of early marriages in some communities. However, mostly the act of ignorance, superstitions, and traditional belief system, plays a role in the lives of those who do marry early. That's why there is no single causative factor that distinctly influences early marriage, rather it is rooted in several socio-cultural and economic factors. Thus, this study revealed that a high majority of respondents have engaged with early marriages. Nevertheless, the occurrence of early marriages is the outcome of prevailing socio-cultural and economic situations displayed via societal values, beliefs, traditions, rituals, and so on. This fact contributes to understanding the prevalence and determinants of early marriage from a wider outlook of the socio-cultural and economic setting. A thorough understanding enables policymakers to improve strategies, that promote higher formal schooling. Which in turn could reduce the prevalence of early marriage and associated beneficial welfare effects in Nepal. Moreover, reducing the socially positive attitude towards early marriage, will no doubt reduce the level of early marriage in Nepal.

**Author Contributions:** Conceptualization, P.C.B., P.K.P. and M.S.; Methodology, P.C.B., P.K.P. and M.S.; Software, M.S., D.R.P. and P.K.P.; Validation, P.C.B., T.P. and S.G.; Formal analysis, P.C.B., P.K.P., D.R.P. and M.S.; Investigation, M.S., P.K.P. and S.G.; Resources, J.I.G., P.K.P., M.S. and D.R.G.; Data collection, M.S. and P.K.P.; Writing—original draft preparation, P.C.B., M.S., P.K.P. and D.R.P.; Writing—review and editing, P.K.P., T.P., M.S. and S.G.; Visualization, P.C.B.; Supervision, P.C.B.; Project administration, M.S. and P.K.P.; Funding acquisition, J.I.G., D.R.G.; and P.C.B. All authors have read and agreed to the published version of the manuscript.

**Funding:** We acknowledge World Vision International Nepal for financial support conducting this study. However, the authors did not receive any fund for the publication purpose.

**Institutional Review Board Statement:** Not applicable.

**Informed Consent Statement:** Informed consent was obtained from all subjects involved in the study.

**Data Availability Statement:** Data will be available on the genuine request to the corresponding author.

**Conflicts of Interest:** The authors declare no conflict of interest.

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
