# Peer review of "Prevalence of Early Marriage and Its Underlying Causes in Nepal: A Mixed Methods Study"

_socsci, doi:10.3390/socsci11040177_

Round 1

Reviewer 1 Report

The article shows a good knowledge of the subject and the literature. Likewise, it presents a good combination of quantitative and qualitative methodologies which strenghten their conclusions. It provides sound evidence and data for a geography that had few studies on the field. For all these reasons I think the article is worth publishing.

Reviewer 3 Report

This a commendable and timely study as concerns about the negative consequences of child marriages are current and being addressed by HR organisations in many Asian and African jurisdictions. 

I have a couple of comments that I feel would strengthen the paper prior to publication.

One, is how you define marriage - an important component missing is 'legal' - essentially marriage is a lawful relationship, not just social, physical and mental.  For this reason, it would be helpful to include what the marriage legislation of Nepal requires. You refer marriage as at to ‘20 years of age’ but clearly there must be legal exemptions to this. Many nations have provisions which allow for younger marriages in certain circumstances eg where both parents, or the court, or for Muslims, a religious Imam or Kadi, give permission/consent. Is this the case in Nepal? Are there penalties eg non-registration or invalidity of marriage where it is under the age of 20 years? Or are there no consequences at all? If so, then strengthening the legal regime including criminal sanctions may be one worthy consideration in your recommendations.

Two, twenty years of age for consent to marriage  is quite high – here in Australia it is 18 years (16 with court approval) and I did wonder about the rationale for 20 years (aspirational and educative perhaps) and I think a few lines on the history and determination of this age for marriage would be contextually helpful.

Three, many casual factors were addressed well. One, not mentioned was pregnancy. In other countries, the social stigma, and legal consequences of  ‘illegitimacy’ for the baby, the mother and entire family (shame and honour) play a part in inducing young marriages. Is there any data on this – link between underage marriage and pregnancy which also plays into religious factors.

Related as well is rape – some nations allow a male to be discharged from a rape charge if he agrees to marry the victim, which in turn has casual links to underage marriage. What is the law in and practice in Nepal?

Four, given the interesting divergence the paper notes with religious factors – it would seem helpful to perhaps elaborate a little on these. For example, Islam require a mahr (dower) and I understand that Hinduism requires a dowery, both of which operate differently but require financial settlement . Does this perhaps explain a difference between Christianity and Buddhism (which have neither – unless it is different in Nepal?)

Five, it was an interesting finding of the paper that the incidence is decreasing. Exploring this a little more would be informative. Does it relate to educational improvement for  women? Increasing GDP? Campaigns?....

Lastly, with English expression I would recommend:

avoiding contractions eg What’s more… to-> furthermore or additionally;

colloquialisms eg ‘married off...to’ -> ‘marry’ so rather than ‘households who married off their sons -> households whose sons marry; & ‘get married earlier’ to ->marry earlier.

Unnecessary or unclear adjectives eg ‘extreme cases of early marriage’.. is this where the child is 14 or under?? Just let the reader know what amounts to extreme; eg highly focuses … what makes this different from focuses?

is how you define marriage - an important component missing is 'legal' - essentially marriage is a lawful relationship. For this reason it would be helpful to include what the marriage legislation actually provides and is it uniform or differences according to religion/ethnicity. 
